# Exploring the Evolving Scope of Neoadjuvant Immunotherapy in NSCLC

**DOI:** 10.3390/cancers14030741

**Published:** 2022-01-31

**Authors:** John F. Roller, Nirmal K. Veeramachaneni, Jun Zhang

**Affiliations:** 1Division of Medical Oncology, Department of Internal Medicine, University of Kansas Medical Center, Kansas City, KS 66205, USA; jroller2@kumc.edu; 2Department of Cardiovascular and Thoracic Surgery, University of Kansas Medical Center, Kansas City, KS 66160, USA; nveeramachaneni@kumc.edu; 3Department of Cancer Biology, University of Kansas Medical Center, Kansas City, KS 66205, USA

**Keywords:** neoadjuvant, immunotherapy, immune checkpoint inhibitors, non-small cell lung cancer, resectable lung cancer

## Abstract

**Simple Summary:**

Immunotherapy with immune checkpoint inhibitors has recently brought a paradigm shift in the treatment of non-small cell lung cancer (NSCLC), but until now, most clinical benefits of immunotherapy have been demonstrated in the setting of advanced or metastatic disease. However, there has been a recent explosion in trial development and research focus exploring whether the benefits of immunotherapy can extend to the neoadjuvant setting for patients with resectable NSCLC. The aim of this review is to thoroughly outline the preclinical rationale for neoadjuvant immunotherapy research. In addition, we summarize and analyze the published interim results as well as results presented at major conferences from the initial early phase trials. An overview of the current and upcoming randomized clinical trials in this field is also provided. Finally, we highlight future challenges and questions that need to be addressed in upcoming research to clarify the role of neoadjuvant immunotherapy in the treatment of NSCLC.

**Abstract:**

While lung cancer remains the leading cause of cancer death worldwide, lung cancer mortality has notably decreased in the past decade. Immunotherapy with immune checkpoint inhibitors have played a noteworthy role in contributing to this improved survival, particularly for patients with non-small cell lung cancer (NSCLC). However, until now the benefits have primarily been seen in patients with advanced or metastatic disease. Several recent early phase and ongoing phase III trials have been assessing whether the treatment benefit of immunotherapy in NSCLC can extend to the neoadjuvant setting for resectable diseases. In this comprehensive narrative review, we evaluate the most recent efficacy and safety data from these studies. We also outline questions that will need to be further examined to legitimate neoadjuvant immunotherapy’s role in NSCLC treatment, including the best surrogate marker of response, the incorporation of liquid biopsy for disease monitoring, the ability to be combined with other treatment modalities, the need for further adjuvant therapy, and potential future treatment combinations.

## 1. Introduction

Lung cancer continues to be the leading cause of cancer-associated death worldwide [1]. In the United States, it is estimated that there will be 235,760 new cases of lung cancer and 131,880 deaths from lung cancer for the year 2021 [2]. The majority of these cases will consist of non-small cell lung cancer (NSCLC) as it is the histologic type for 80–85% of lung cancer cases in the United States [3,4]. Recent progress in the research of precipitating factors, screening, and treatment for lung cancer has led to a notable decrease in lung cancer mortality in the past decade [2]. Immunotherapy with immune checkpoint inhibitors has been a noteworthy contributor to this improved survival rate, particularly with NSCLC. The current areas where the addition of immunotherapy in NSCLC treatment has become an integral component of standard care have been in the advanced and metastatic settings. However, researchers now have begun postulating that the clinical benefits of immunotherapy can potentially extend to patients with earlier stages of NSCLC. As a result, an eruption of recent early phase and ongoing phase III trials evaluating neoadjuvant immunotherapy in resectable NSCLC has been seen. In this review, we evaluate the most recent efficacy and safety data from these up-to-date studies. We also speculate the future of neoadjuvant immunotherapy for NSCLC by identifying potential challenges to its development, questions needed to be further examined to legitimate its role in NSCLC care, and potential treatment combinations that could maximize its clinical efficacy.

## 2. Materials and Methods

The references to complete this review were identified by literature search utilizing databases including PubMed, Google Scholar, and Clinical Key. Abstracts from major clinical conferences including the International Association for the Study of Lung Cancer (IASLC) World Conference on Lung Cancer, American Association for Cancer Research, European Society for Medical Oncology, and American Society of Clinical Oncology were also included. Included articles and abstracts were from database inception to 1 November 2021. The literature search was performed using the following search terms: (1) neoadjuvant, (2) immunotherapy, (3) resectable lung cancer, (4) non-small cell lung cancer, (5) checkpoint inhibitors. Only articles and abstracts published in English were included in this review. The results of various trials and studies accordingly were summarized narratively.

## 3. Neoadjuvant Therapy in Oncology

Neoadjuvant therapy is a treatment approach that has been practiced in oncologic care for over fifty years. It comprises any form of treatment—chemotherapy, radiation therapy, endocrine therapy, or immunotherapy—that is given for cancer before the definitive treatment with curative intent, usually surgery, is conducted. A key advantage for neoadjuvant therapy is providing treatment at the earliest opportunity to minimize the chance for micrometastases to grow when patients receive and recover from local therapy [5]. It also allows for an assessment of systemic therapy effects on the visible disease, and it is intended to decrease the extent of tumor involvement leading to less complex and morbid surgical procedures. It may be given more reliably than adjuvant therapy due to a patient’s potential for complications from primary treatment making additional treatment difficult [5,6]. On the other hand, there are several risks to neoadjuvant therapy including obscuring pathologic staging, delaying the start of curative treatment, and increasing treatment toxicity, which could also lead to more complex surgical fields [7].

As early as 1956, neoadjuvant therapy was proposed for the treatment of choriocarcinoma [8]. Perioperative chemotherapy was then introduced as a potential component of breast cancer treatment in 1968 [9]. Subsequently, neoadjuvant therapy has developed an established role in the downstaging of several malignancies such as rectal, head and neck, breast, esophageal, colon, bladder, and lung cancer [7,10,11,12,13,14]. In addition, by allowing for the assessment of major pathological response (MPR, defined as less than 10% residual tumor) and pathologic complete response (pCR) at time of surgery, neoadjuvant therapy has also provided important prognostic information on long-term clinical outcomes. MPR and pCR have been seen to be indicators or surrogates of improved overall survival in patients with various malignancies such as breast or esophageal cancer, as well as lung cancer [15,16,17].

## 4. Neoadjuvant Immunotherapy in Oncology

Immunotherapies with immune checkpoint inhibitors targeting the interaction of programmed death 1 (PD-1) with its ligands PD-L1 and PD-L2 or the cytotoxic T lymphocyte–associated protein 4 (CTLA-4) pathway, have recently become vital treatment components for a variety of malignancies. While immunotherapy’s initial key clinical benefits and tumor activity were seen in the metastatic setting, burgeoning research attention is now being placed in exploring these agents for benefits in the neoadjuvant setting.

The preclinical rationale for evaluating neoadjuvant immunotherapy is varied. Neoadjuvant chemotherapy is thought to primarily decrease or debulk tumors preoperatively, but neoadjuvant immunotherapy may better target and eliminate micrometastatic disease and the subsequent chance of recurrence by enhancing systemic immunity against tumor antigens [7,18]. There are two key models by which immune checkpoint inhibitors are thought to promote antitumor immunity by primarily targeting micrometastatic disease. The first suggests anti–PD-(L)1 therapies rejuvenate tumor-specific cytotoxic T cells already residing in the tumor microenvironment, leading to their activation, proliferation, and trafficking to micrometastatic deposits. Second, dendritic cells’ presentation of tumor antigens to T-cells in tumor-draining lymph nodes appears to be the focal point for anti–PD-(L)1 activity. These tumor-specific T cells can then enter the bloodstream and migrate to tumor sites [18] (Figure 1). Neoadjuvant use of immunotherapy is also believed to be an advantageous approach because the tumor cell destruction by immunotherapy requires antigen presentation by the tumor cells to be recognized by host T-cells. Therefore, employing immunotherapy while the primary tumor is intact provides higher levels of available tumor antigen present to enhance T-cell priming compared to postresection adjuvant therapy [7,18].

These hypotheses have been tested in several preclinical studies. Liu et al. utilized two spontaneously metastasizing transplantable mouse breast cancer models, in which various combination immunotherapies were administered in either a neoadjuvant or adjuvant setting before or after primary tumor resection [19]. They subsequently compared the efficacy of neoadjuvant and adjuvant anti–PD-1 therapy after primary tumor resection. Survival was noted to be significantly longer with neoadjuvant versus adjuvant immunotherapy, even when treatment was introduced at the same time after tumor implantation. Additionally, this same model found that neoadjuvant chemotherapy with paclitaxel did not improve overall long-term survival when compared with adjuvant chemotherapy. The observed improvement in survival with immunotherapy combinations was seen to be associated with greater numbers of tumor-specific CD8+ T-cells in the lungs and blood of the mice, as well as enhanced cytokine production in response to an endogenous retroviral antigen expressed by tumor cells.

Consequently, the authors suggested blood sampling for tumor-specific CD8+ T-cells before and after surgery may provide a predictive biomarker from neoadjuvant immunotherapy [7,18,19]. Other groups have subsequently tested the efficacy of various neoadjuvant immunotherapy combinations in different preclinical mouse models. These studies further supported the hypothesis that combined immunotherapy effected a greater reduction in micrometastatic disease and improved survival when given in a neoadjuvant setting compared to adjuvant treatment [20,21,22]. Based on the promising results of these pre-clinical studies, the neoadjuvant paradigm has become a focus of investigation in clinical settings.

## 5. Neoadjuvant Immunotherapy in NSCLC

NSCLC is a particular malignancy that has seen an explosion of recent trials as well as ongoing trials evaluating neoadjuvant immunotherapy whether alone or in combination with chemotherapy. Forde et al. conducted the first of these early phase trials [23]. This single-arm study with 22 enrolled patients (21 included in study analysis) analyzed the administration of two doses of neoadjuvant nivolumab every two weeks for patients with stage I to IIIA resectable NSCLC. Twenty of the twenty-two enrolled patients received the two planned doses of nivolumab. During evaluation of the study’s primary end points of safety and feasibility, 5 of 22 patients (23%) experienced treatment-related adverse events of any grade, and only 1 patient experienced an event grade 3 or higher (pneumonia). No treatment-related surgical delays were observed, and the median interval between administration of the second dose of nivolumab and surgery was 18 days. In addition, the secondary end point of a major pathologic response (MPR), which was defined as up to 10% viable tumor cells at surgery, occurred in 9 of the 20 patients (45%; 95% confidence interval [CI]: 23–68%) with 2 patients (10%) having a complete pathologic response in the primary tumor. Another key finding from the study is that it supported pre-clinical data results that nivolumab administration leads to substantial expansion of CD8 cells in the resected tumor as well as peripheral blood after surgery. This expansion correlated with the therapy response.

Gao et al. conducted a single-center, single-arm, phase 1b study that evaluated the administration of two doses of weekly sintilimab, a PD-1 inhibitor approved in the People’s Republic of China, as neoadjuvant treatment in 40 enrolled patients with stage IA to IIIB NSCLC [24]. Primary endpoints were adverse events, operative complications, and nonoperation delay rate (≤43 days from first dose of sintilimab until operation). All 40 patients received two doses of sintilimab, with 37 (92.5%) receiving surgery. Twenty-one patients (52.5%) experienced immunotherapy-related adverse events. Three adverse events grade ≥3 (7.5%) were seen with two of these being immunotherapy-related. Only two patients (5%) had a delay in surgery. Of the 37 patients who underwent surgery, 15 patients (40.5%, 95% CI: 24.8–57.9%) achieved an MPR and six patients (16.2%, 95% CI: 6.2–32.0%) achieved a pCR. Sintilimab was also provided as adjuvant treatment for 15 patients (40.5%).

The Lung Cancer Mutation Consortium 3 (LCMC3) trial is the largest monotherapy trial of immunotherapy in resectable NSCLC [25,26]. It is a phase 2 trial analyzing two cycles of neoadjuvant atezolizumab followed by resection 30 to 50 days from the first cycle in patients with Stage IB to IIIB NSCLC and no targetable mutations. Patients benefitting from neoadjuvant treatment could continue adjuvant atezolizumab for 12 months. One hundred and eighty-one patients were enrolled in the study with 159 patients (88%) receiving surgical resection, 155 with pathologic stage evaluations, and 144 included in the efficacy analysis. Stage IIIA comprised the highest proportion of enrolled patients at 40%, followed by Stage IIB at 33%. Stage IIIB was seen in only 8% of patients evaluated. The median time from the end of neoadjuvant therapy to surgery was 22 days, which is a shorter window than that of many neoadjuvant chemotherapy trials [27]. The primary endpoint was MPR. This was achieved in 21% of patients in the trial with 7% of patients achieving a complete pathologic response, meeting the pre-determined primary endpoint. Additionally, 43% of patients had their disease downstaged after atezolizumab treatment, while 19% had their disease upstaged. Complete or R0 resection was achieved in 92% of patients. A safety analysis showed toxicities similar to previous studies. Treatment related adverse events (TRAEs) of any grade were seen in 56% of patients while only 5% received a grade 3 event or higher. Immunotherapy related adverse events (irAEs) were only seen in 24.3% of patients, with 2.2% experiencing grade 3 or higher. These interim results were presented at the 2020 World Conference on Lung Cancer [26]. Future planned analysis will present data on DFS and OS at 1 year and 18 months.

The first interim analysis for the NEOMUN trial were also recently reported [28]. This investigator initiated open-label, single arm, prospective, single-center, phase II trial evaluated patients with biopsy-proven, treatment-naive, clinical stage II-IIIA resectable NSCLC. Enrolled patients were assigned to receive a fixed dose of pembrolizumab every 3 weeks for 2 cycles prior to surgical treatment, and the planned interim analysis was conducted after the first 15 patients were enrolled and treated. Primary objectives of this trial were to assess the safety and feasibility of neoadjuvant immunotherapy, as well as evaluating the clinical and pathologic tumor response. Twelve patients (80%) completed the planned treatment with pembrolizumab and surgery per protocol. Two patients did not receive a second dose of pembrolizumab due to grade 3 adverse events, and 1 patient had a delay in surgery due to TRAEs. Five patients enrolled (33%) were seen to have clinically relevant TRAEs. Two patients were seen to have MPR and 2 other patients had a pCR (27% with at least MPR).

A neoadjuvant immunotherapy trial that was recently presented at the ESMO Virtual Congress 2020 was the IONESCO trial. This phase II trial analyzed the administration of neoadjuvant durvalumab, 750 mg, for 3 courses (days 1, 15, 29) prior to surgical resection, which followed between 2 to 14 days after their last durvalumab infusion, in patients with stage IB (tumor > 4 cm)-IIIA resectable NSCLC. Fifty patients were enrolled in the study with 46 receiving surgical resection. The primary endpoint was the rate of complete surgical resection (R0), with secondary endpoints including MPR, overall survival (OS), and disease-free survival (DFS). Forty-one patients (90%) were seen to achieve R0, with MPR 18.6% and pCR 7%, but enrollment for this study was stopped prematurely due excess in 90-day postoperative mortality with 4 deaths (9%) observed. However, the authors of the study did not attribute any of these deaths to durvalumab toxicities, but rather they were thought to be due to pre-existing comorbidities highlighting the importance of caution with comorbidities in neoadjuvant immunotherapy trials [29].

The ESMO Virtual Congress 2020 also showcased the interim results of the PRINCEPS trial [30]. In this phase II trial, one dose of neoadjuvant atezolizumab, 1200 mg, was given prior to surgical resection in patients with clinical stage IA (≥2 cm)-IIIA NSCLC. Surgical resection was planned between day 21 and day 28 of treatment. The primary endpoint was the rate of patients without major toxicities or morbidities from day 1 until 1 month after the surgery. MPR was also assessed. Among the 30 enrolled patients, all patients had their planned surgery with none being delayed by >15 days. Twenty-nine had R0 resection and only 3 patients had surgical complications, with no grade 5 toxicity seen. No pCR was seen; however, MPR was reported in 4 patients (14%) and a pathological response ≥ 50 (less than 50% residual tumor cells) was reported in 12 patients (41%). In addition, a metabolic response (18F-FDG PET/CT, variation of SUVmax) was also noticed.

Above mentioned neoadjuvant studies using single agent anti-PD-1/L1 are summarized in Table 1 and Table 2.

## 6. Combination Neoadjuvant Immunotherapy in NSCLC

To further explore the potential for neoadjuvant immunotherapy in NSCLC, trials are being conducted that analyze the effect of immune checkpoint inhibitor combinations. The NEOSTAR study is a Phase II randomized clinical trial that evaluated two parallel, randomized arms, with arm A utilizing neoadjuvant nivolumab (3 mg/kg IV every 14 days on D1, D15, and D29) and arm B utilizing a combination of neoadjuvant nivolumab and ipilimumab (nivolumab 3 mg/kg IV every 14 days on D1, D15 and D29 and ipilimumab 1 mg/kg IV on D1 only) for resectable stage IA to IIIA NSCLC [31]. This single-institution study enrolled 44 patients, and 23 patients were randomized to nivolumab alone while 21 patients received nivolumab plus ipilimumab. Forty-one of these patients completed the planned neoadjuvant therapy, and 37 patients had surgery on trial while 2 patients underwent surgery off-trial after received additional therapies. The primary endpoint was MPR. The predetermined level for which the study was considered to meet this endpoint was ≥6 MPRs in 21 evaluable patients for each arm, based on historical controls for neoadjuvant platinum-based chemotherapy that showed a response rate of approximately 15% [32]. The median time to surgery was 31 days after the last dose of nivolumab. For the 37 patients who received surgery on trial, 8 of 16 patients (50%, 95% CI = 25–75%) who received dual therapy achieved MPR while only 5 of 21 patients (24% 95% CI = 8–47%) who received nivolumab achieved MPR. In addition, a higher rate of pathologic complete response was seen with dual therapy compared to nivolumab alone, 38% vs. 10%. Consequently, only the dual therapy was seen to meet the predetermined primary end point, suggesting that combination neoadjuvant immunotherapy may bring superior results compared to single-agent immunotherapy.

Reuss et al., also performed a multicenter, open-label, single-arm phase Ib/II study evaluating combination immunotherapy for patients with resectable Stage IB (≥4 cm)–IIIA NSCLC [33]. Enrolled patients received nivolumab, 3 mg/kg IV, in combination with ipilimumab, 1 mg/kg IV, 6 weeks prior to planned resection. They then received two additional doses of nivolumab, 3 mg/kg, at approximately four and two weeks preoperatively. Primary endpoints were feasibility and safety, with feasibility defined as a delay in surgery of ≤24 days from the preplanned surgery date, and safety defined by adverse events according to common terminology criteria for adverse events (CTCAE) V.4.0. Pathologic response was a key exploratory endpoint. Nine patients were enrolled in the trial with all enrolled patients being deemed fit for planned surgery without treatment-related delays, meeting the criteria for feasibility. Four patients were seen to have progressive disease by the response evaluation criteria in solid tumors (RECIST) prior to surgery. One of the patients still underwent surgical resection leading to a total of six patients within the trial receiving surgery. pCR was observed in two of six (33%) resected tumors. Six of the nine patients (67%) experienced any grade of TRAEs, and three (33%) experienced grade ≥3 TRAEs. The main aspects of these combination neoadjuvant immunotherapy trials are summarized in Table 3.

## 7. Combining Neoadjuvant Immunotherapy with Other Modalities and Adjuvant Use

Based off the initial promising results from neoadjuvant immunotherapy trials in NSCLC, researchers have begun developing trials to examine whether the benefits of immunotherapy can be enhanced through its concurrent use with other neoadjuvant modalities such as chemotherapy, radiotherapy, or novel agents. Neoadjuvant chemoimmunotherapy is a combined treatment approach that has several current reported results, which are outlined in Table 3.

Upon review, it appears that the combination of neoadjuvant immunotherapy and chemotherapy may lead to increased antitumor activity as increased MPR and pCR rates, ranging from 27–86% and 9–63%, respectively, are seen in most of these trials in comparison to MPR and pCR rates with solely immunotherapy, ranging from 12–50% and 0–38% [23,24,25,26,27,28,29,30,31,32,33,34,35,36,37,38,39,40,41,42,43,44]. On the other hand, it suggests that immunochemotherapy may not be as well tolerated as immunotherapy alone as, on the whole, higher rates of AEs and trAEs are seen with these treatments. However, as these trials are mainly phase II trials with no controls or comparators and had significant heterogeneity in their design, comparison across these trials cannot lead to sound conclusions. Instead, it indicates that further prospective randomized trials are needed to determine which neoadjuvant approach in which specific patient population will lead to higher clinical efficacy and survival.

One such current phase III trial evaluating between neoadjuvant immunotherapy and neoadjuvant chemoimmunotherapy is the Checkmate-816 trial (NCT02998528). While initial reported results from this trial focused on evaluating between neoadjuvant nivolumab plus platinum-doublet chemotherapy and neoadjuvant platinum-doublet alone, another experimental arm of this study is looking at neoadjuvant combination immunotherapy with nivolumab plus ipilimumab. These initial reported results showed that the study’s first primary endpoint was achieved as the addition of nivolumab to a platinum-doublet chemotherapy preoperatively produced a decisively significant increased pCR rate compared to platinum doublet chemotherapy alone, in the intention to treat the population (ITT) at 24.0% vs. 2.2% (odds ratio 13.94 [99% CI 3.49–55.75]; *p* < 0.0001). The observed improvement in pCR was observed across all major subgroups including disease stage (IB/II [26.2% vs. 4.8%]; ≥IIIA [23.0% vs. 0.9%]) and PD-L1 status (<1% [16.7% vs. 2.6%]; ≥1% [32.6% vs. 2.2%]). MPR rates were also significantly improved with nivolumab + chemotherapy (36.9% vs. 8.9%) [38,39].

Radiotherapy is another treatment modality that is suggested and currently being evaluated to be a potent partner to immunotherapy in the neoadjuvant setting. This combination is believed to be potentially efficacious as immunotherapy has been hypothesized to accentuate the abscopal effect of radiotherapy, which is its ability to generate immune-mediated anti-tumor effects leading to deterioration of non-irradiated metastases that are distant from the primary irradiated site [45,46,47].

Altorki et al. recently reported data of a single-center, randomized, controlled, phase 2 trial, comparing neoadjuvant durvalumab alone with neoadjuvant durvalumab plus stereotactic radiotherapy in patients with resectable NSCLC [48]. Sixty enrolled patients (30 in each arm) with stage I-IIIA NSCLC were randomly assigned 1:1 to receive either 2 cycles of neoadjuvant durvalumab monotherapy at a dose of 1.12 g IV, or 2 cycles of neoadjuvant durvalumab plus stereotactic body radiotherapy (8 Gy × 3 fractions). In the absence of systemic disease progression, surgical exploration was performed within 2–6 weeks following the second cycle of durvalumab. The primary endpoint was a rate of MPR. Secondary endpoints were 2-year disease-free survival for the whole cohort compared with historical controls and the difference between both groups of the trial in radiographic response and safety of neoadjuvant therapy. Results from the study showed a notable increased rate of MPR for neoadjuvant immunotherapy with radiotherapy as 16 of 30 patients (53.3%, 95% CI 34.3–71.7%) achieved MPR in this group compared to just 2 of 30 patients (6.7%, 95% CI 0·8–22.1%) who received durvalumab alone. Eight of the 16 patients who achieved MPR in the combined group were seen to achieve pCR. Grade 3–4 adverse events occurred in five of 30 patients (17%) in the durvalumab monotherapy group and six of 30 (20%) in the durvalumab plus radiotherapy group. Table 4 outlines several other current trials evaluating neoadjuvant radiotherapy with immunotherapy alone or in combination with chemotherapy.

Another notable clinical question regarding neoadjuvant immunotherapy treatment is whether further immunotherapy in the adjuvant setting is needed, especially for those patients who have achieved pCR. It has not yet been determined if further adjuvant immunotherapy helps better eradicate micrometastatic disease or simply leads to increased treatment toxicity. Most of the major current phase III trials evaluating neoadjuvant chemoimmunotherapy, as outlined in Table 5, are incorporating the addition of adjuvant immunotherapy for around 1 year. Accordingly, more insight into these queries is eagerly awaited.

## 8. Analyzing End Points for Neoadjuvant Therapy in NSCLC

In the clinical evaluation of neoadjuvant therapy, the ultimate goal is to demonstrate improved survival over that attained by adjuvant therapy. Consequently, overall survival (OS) should be the vital end point in neoadjuvant trials including those with neoadjuvant immunotherapy. However, determining OS can be a quite lengthy and difficult process, especially for neoadjuvant treatment, due to the amount of observed time and the large number of patients needed to ascertain differences in survival. These problems can lead to delays in conducting later stage trials, as well as the development of new therapies. Thus, researchers have recognized the significance of identifying surrogate markers that can be determined in a timely fashion, as well as accurately predict OS.

Disease free survival (DFS) has been one such surrogate marker that has been utilized for other malignancies, such as breast cancer or colorectal cancer [49,50]. However, DFS still is seen to require an undesirable amount of time to determine to enable efficient drug development. Radiologically ascertained tumor shrinkage measured as an objective response rate (ORR) has been another studied surrogate marker. However, in several studies, including a meta-analysis of 14 trials of advanced NSCLC by Blumenthal et al. [51], ORR was seen to not be a reliable surrogate of OS in NSCLC [52,53,54].

As a result, evaluations of pathologic responses to neoadjuvant therapy have been the focus of most recent trials as they can be discerned at the time of surgery. pCR, defined by the absence of residual invasive and in situ cancer in a resected specimen and all sampled lymph nodes on hematoxylin and eosin evaluation after neoadjuvant systemic treatment, has been seen as appropriate surrogate marker for OS in NSCLC [32,55,56]. Utilizing pCR for neoadjuvant therapy trials has proven difficult, however, as low rates of pCR at around 10% or less have been observed in these trials. These low rates complicate the ability to assess whether this endpoint can be reached. Consequently, MPR, defined as ≤10% of viable tumor cells in a surgically resected specimen, has been utilized more frequently as it has been achieved at higher rates.

Several studies have suggested that MPR after neoadjuvant therapy may reliably predict OS in NSCLC [17,57,58,59]. However, these studies have mainly been retrospective in nature focusing on chemotherapy, and MPR as a surrogate marker still needs to be validated in prospective trials as well as in use with neoadjuvant immunotherapy. In addition, there are several potential impediments or limitations to use of MPR that need to be further considered and evaluated. First, there is concern for the ability to precisely determine MPR due to inherent interobserver variability amongst pathologists. The International Association for the Study of Lung Cancer (IASLC) has tried to minimize this imprecision by publishing multidisciplinary recommendations to standardize the processing of resection specimens of lung cancer as well as defining pathologic responses to systemic therapies [60]. Another concern is the fact more trials are analyzing adjuvant therapy in combination with neoadjuvant therapy which could confound the correlation between MPR and survival event rates [61]. Finally, focus on MPR may not consider biological mechanisms produced by neoadjuvant immunotherapy, such as the peripheral expansion of tumor-specific T cells, that could influence OS. Therefore, mechanism-focused biomarkers may prove to be important evaluation surrogates for OS in future trials.

## 9. Role of Circulating Tumor DNA and Immune Biomarkers in Neoadjuvant Immunotherapy Assessments

To further consider the potential for mechanism-focused biomarkers as surrogate end points, recent research has focused on analyzing circulating tumor DNA (ctDNA) though liquid biopsy as a future predictor for recurrence and OS. ctDNA is DNA released from damaged or apoptotic tumor cells present in blood plasma. There are several malignancies, in particular colorectal cancer and hematological malignancies, that have an observed presence of ctDNA, correlating with an increased risk of recurrence [62,63,64]. However, these studies have mainly been in the adjuvant setting. Recently, Romero et al. presented an exploratory analysis of the phase II NADIM trial evaluating pretreatment ctDNA as a predictor of long-term survival at the International Association for the Study of Lung Cancer (IASLC) 2021 World Conference on Lung Cancer [65]. Amongst the 46 patients included in the trial, 43 pretreatment plasma samples after neoadjuvant immunotherapy were available for ctDNA analysis. ctDNA was analyzed by next-generation sequencing, using the Oncomine Pan-Cancer Cell-Free Assay. With a median follow-up of 38 months, multivariate analysis showed that patients with low ctDNA levels (<1% mutant allele frequency) in the pretreatment sample had significantly better progression-free survival (hazard ratio [HR] = 0.22) and overall survival (HR = 0.04) than patients without. Adjusted C-statistic (c) was used to predict progression-free survival for ctDNA at 0.68, which was superior to that of the RECIST criteria (c = 0.61) and similar to that of the pathologic response. While this was just an exploratory analysis, the authors propose the results indicate ctDNA should be considered for the primary endpoint in future NSCLC neoadjuvant trials [65].

Checkmate-816 is another recent prospective trial that evaluated the potential relationship between ctDNA and clinical response. In this study, ctDNA analyses were performed on plasma samples collected on day 1 before each of the 3 planned treatment cycles. The rate of ctDNA clearance was the primary component examined, which was defined as change from detectable levels of ctDNA at cycle 1 to undetectable ctDNA at cycle 3. An exploratory subset analysis revealed a notably higher ctDNA clearance rate was seen in the nivolumab plus chemotherapy group versus chemotherapy alone, at 56% to 34%. This corresponded with the increased pCR and objective response rate observed with nivolumab plus chemotherapy [38].

The results of the Checkmate-816 trial appear to further demonstrate the need for additional research focused on evaluating ctDNA as a potential response biomarker for neoadjuvant immunotherapy. It may also be important to look into the dynamic change of ctDNA at different time points during and after neoadjuvant treatment. In addition, the potential value of integrating immune biomarkers such as peripheral blood and/or intratumoral immune cell repertoire and cytokine levels, as well as gut and intratumoral microbiomes, may also need to be explored.

## 10. Discussion

Immunotherapy with immune checkpoint inhibitors has proven to be a novel integral component of NSCLC care in recent years. To further identify the scope of immunotherapy’s clinical utility, a significant number of recent and current trials are exploring if immunotherapy’s benefits can extend to the neoadjuvant setting for resectable NSCLC. The initial results of preclinical and early phase trials suggest further research focus to this treatment approach is warranted by exhibiting the safety, feasibility, and versatility of neoadjuvant immunotherapy. Neoadjuvant immunotherapy additionally has been seen to be a rather potent tumor debulking agent in these studies with MPRs ranging up to 45% when used alone, and even higher, up to around 80–85%, when used in conjunction with chemotherapy.

As more and more data have recently been reported from neoadjuvant immunotherapy trials, it will be important to analyze the major trends and associations within these results to better identify future research needs. Cao et al. recently helped address this need by conducting a systemic review and meta-analysis of sixteen studies with eighteen publications evaluating neoadjuvant immunotherapy in resectable NSCLC [66]. The review included all trials with data published up to August 2021. Five hundred and forty-eight patients who had received at least one cycle of neoadjuvant immunotherapy, with 507 patients undergoing subsequent surgery, were included in the analysis. This review found that 52% (95% CI: 42–62%, I^2^ = 73%) of patients who underwent surgery following neoadjuvant immunotherapy achieved MPR, 24% (95% CI, 17–34%; I^2^ = 76%) achieved pCR of the primary lesion, and 20% (95% CI: 9–36%; I = 86%) achieved pCR of both the primary lesion as well as sampled lymph nodes. With this finding, Cao et al., prudently recognized that there were discrepancies in how trials reported endpoints of MPR and pCR, with only some of the trials addressing whether the resected lymph nodes were also assessed. The variations in endpoint reporting from these trials compromises the ability to assess clinical outcomes from neoadjuvant immunotherapy and should be standardized in future studies.

Another key discovery from the meta-analysis is that neoadjuvant immunotherapy appears to be safe and feasible, as 96% of patients received intended surgery after systemic neoadjuvant treatment, mortality within 30 days of surgical resection was 0.6%, and only 2.0% of patients were delayed from their intended time of operation after treatment with neoadjuvant immunotherapy. Surgical morbidities within the analysis were also seen to be similar to studies evaluating thoracic resections without neoadjuvant immunotherapy [66,67].

Several crucial questions and challenges remain that still need to be addressed before neoadjuvant immunotherapy can be regarded as a recommended treatment strategy. The ideal administration of neoadjuvant immunotherapy is still yet to be determined, including the optimal patient population for the consideration of a single agent immune checkpoint inhibitor vs. combination therapy, the number of preoperative cycles needed, and timing with respect to surgery. Surgical procedures have been noted to be more challenging after neoadjuvant immunotherapy, with increased adhesions and fibrosis in surgical fields observed [7,44]. Thus, identifying how to deliver neoadjuvant immunotherapy in a way which minimizes these potential surgical complications will be imperative. In addition, delineating which patients would benefit most from this therapy with minimal toxicity is needed as early neoadjuvant chemoimmunotherapy trials have shown noteworthy rates of adverse events. One potential approach to identify these patients is the use of predictive biomarkers to direct selection of a particular therapy. Another key question is determining whether adjuvant or consolidative treatments are necessary. Finally, there still needs to be insight in recognizing the best surrogate marker or endpoint for long-term survival to establish clinical efficacy and relevancy for neoadjuvant immunotherapy.

A review of the initial trials with neoadjuvant immunotherapy highlights several limitations of these studies that need to be considered and addressed in future trials. There is much heterogeneity in the patient populations, trial designs, types of immunotherapy provided, and endpoints evaluated in early phase trials within this field. This heterogeneity limits the ability to compare results across trials and develop improved future studies. More collaboration is needed amongst investigators to identify common standards necessary for trials to enable greater focus on their intended variables and increase external validity.

The significant heterogeneity of these trials also limits the ability to form reliable conclusions in this review. The risk of biases in this review is recognized, particularly publication bias as most of the studies included were those which reported preliminary results. Therefore, revisions and updates to this review will be needed as further data from these trials are published.

## 11. Conclusions

Despite the challenges and limitations noted, the promising results from the early clinical trials with neoadjuvant immunotherapy demonstrate that it is a safe and feasible approach that merits further study. The use of neoadjuvant immunotherapy alone or in combination with other therapies may soon revolutionize the standard of care for resectable NSCLC, such as the Checkmate 816 regimen. Consequently, the results of ongoing randomized trials to determine if and how this novel approach can be utilized optimally to treat NSCLC patients are eagerly awaited.

## Figures and Tables

**Figure 1 cancers-14-00741-f001:**
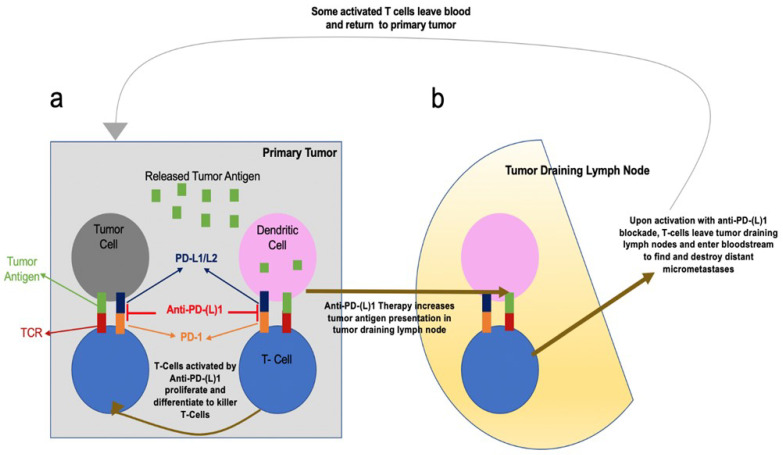
Main theories for enhanced systemic antitumor T-cell activity by neoadjuvant anti-PD-(L)1 Therapy. (**a**) Anti-PD-(L)1 therapy can stimulate and rejuvenate tumor-specific cytotoxic T cells residing in the tumor microenvironment. Thus, creating an in-situ expansion of tumor-specific T-cells by facilitating their activation, proliferation, and trafficking to micrometastatic deposits. (**b**) Anti-PD-(L)1 therapy can also improve stimulation of tumor-specific T cells by tumor antigen-presenting dendritic cells within tumor draining lymph nodes. From [18]. Reprinted with permission from AAAS.

**Table 1 cancers-14-00741-t001:** Summary of Efficacy Outcomes from Neaodjuvant Single Agent Immunotherapy Trials with Reported Results.

Study	NCT Number	Population No. in Study (Stage I/II/III)	Histologic Type: *n* (%)	Study Arm	Primary Endpoints	% of Complete Surgical Resection (R0)	MPR and pCR Rates, *n*/*n* with Resection (%)
Forde et al. (2018) [23] Phase II	02259621	Stage I–IIIA Resectable NSCLC, 21 (4/10/7)	Adeno: 13 (62) SCCa: 6 (29) Other: 2 (10)	2 cycles (3 mg/kg every 2 weeks) nivolumab pre-operatively	Safety and Feasibility	95%	MPR: 9/20 (45) Stage I/II/III: 2/5/2 pCR: 2/20 (10) Stage I/II/III: 0/1/1
Gao et al. (2020) [24] Phase Ib	17013726	Stage I (≥2 cm)–IIIB Resectable NSCLC, 40 (8/14/18)	Adeno: 6 (15) SCCa: 33 (82.5) Other: 1(2.5)	2 cycles of sintilimab (200 mg every 3 weeks) pre-operatively	Safety and Feasibility	97%	MPR: 15/37 (40.5) Stage I/II/III: 1/5/9 pCR: 6/37 (16) Stage I/II/III: 0/2/4
LCMC3 (2021) [25,26] Phase II	02927301	Stage IB–IIIB Resectable NSCLC, 181 (16/80/85)	Non-squamous: 112 (62) Squmous: 69 (38)	2 cycles of atezolizumab (1200 mg every 2 weeks) pre-operatively	MPR	91%	MPR: 30/147 (20) Stage I/II/III: 3/12/15 pCR: 10/147 (7) (Patients without EGFR/ALK mutations who underwent surgery)
NEOMUN (2020) [28] Phase II	03197467	Stage II–IIIA Resectable NSCLC, 15 (6/9)	Adeno: 13 (87) SCCa: 2 (13)	2 cycles of pembrolizumab (200 mg every 3 weeks) preo-peratively	Safety and Feasibility	100%	MPR: 4/15 (27) Stage I/II/III: 0/2/2 pCR: 2/15 (13) Stage I/II/III: 0/2/0
IONESCO (2020) [29] Phase II	03030131	Stage IB (≥4 cm)–IIIA (non N2) Resectable NSCLC, 46 (5/13/28)	Adeno: 23 (50) SCCa: 19 (41) Other: 4 (9)	3 cycles of durvalumab (750 mg every 2 weeks) preo-peratively	% of complete surgical resection (R0)	90%	MPR: 8/43 (18.6) pCR: 3/43 (7)
PRINCEPS(2020) [30] Phase II	02994576	Stage I (≥2 cm)–IIIB (non N2) Resectable NSCLC, 30 (15/6/9)	Adeno: 25 (83) Other: 5 (17)	1 cycle of atezolizumab (1200 mg) pre-operatively	Safety and Feasibility	96.7%	MPR: 4/29 (14) pCR: 0/29 (0)

MPR = Major pathologic response; pCR = Pathologic complete response. If there is no further breakdown into stages of NSCLC within the table, it means there is no corresponding reported data.

**Table 2 cancers-14-00741-t002:** Summary of Safety Outcomes from Neaodjuvant Single Agent Immunotherapy Trials with Reported Results.

Study	Population, *n* in Study (Stage I/II/III)	Study Arm	Protocol-Specified Timeframe for Surgery	Pts with Surgery (%)/Pts Received Neoadjuvant Therapy	Median Time to Resection	30-Day Post-Operative Mortality/90 Day Post-Operative Mortality	tRAEs (%)
Forde et al. (2018) [23] Phase II	Stage I–IIIA Resectable NSCLC, 21 (4/10/7)	2 cycles (3 mg/kg every 2 weeks) nivolumab pre-operatively	Approximately 4 weeks after the first dose of Nivolumab	20 (95%)/21	18 days No treatment-related delays	0%/0% (One patient without recurrence died from a traumatic head injury that was unrelated to the study treatment)	trAEs Any Grade: 5/22 (23) trAEs ≥ G3: 1/22 (4.5)
Gao et al. (2020) [24] Phase Ib	Stage I (≥2 cm)–IIIB Resectable NSCLC, 40 (8/14/18)	2 cycles of sintilimab (200 mg every 3 weeks) pre-operatively	29–43 days after first dose of sintilimab	37 (92.5%)/40	NR 2/37 (5%) had treatment-related delays	5%/NA	trAEs Any Grade: 21/30 (52.5) trAEs ≥ G3: 4/30 (10)
LCMC3 (2021) [25,26] Phase II	Stage IB–IIIB Resectable NSCLC, 181 (16/80/85)	2 cycles of atezolizumab (1200 mg every 2 weeks) preop-eratively	Day 40 ± 10 d after first dose of atezolizumab	159 (88%)/181	NR 19/159 (12%) had treatment-related delays	0.6%/0.6%	Preoperative trAEs Any Grade: 101/181 (56) trAEs ≥ G3: 9/181 (5) PostoperativetrAEs Any Grade: 57/159 (36) trAEs ≥ G3: 21/159 (14)
NEOMUN (2020) [28] Phase II	Stage II–IIIA Resectable NSCLC, 15 (6/9)	2 cycles of pembro-lizumab (200 mg every 3 weeks) preoperatively	1–3 weeks after last cycle of pembro-lizumab	13 (87%)/15	NR 1/13 (8%) had treatment-related delays	0%/NA	trAEs Any Grade: 8/15 (53) trAEs ≥ G3: 5/15 (33)
IONESCO (2020) [29] Phase II	Stage IB (≥4 cm)–IIIA (non N2) Resectable NSCLC, 46 (5/13/28)	3 cycles of durvalumab (750 mg every 2 weeks) preoperatively	Between day 2 and 14 after last cycle of durvalumab	46 (100%)/46	37 days	NA/9% (Study was stopped due to excess in 90-day postoperative mortality)	None Reported. No episode of 90-day postoperative mortality thought to be treatment related.
PRINCEPS (2020) [30] Phase II	Stage I (≥2 cm)–IIIB (non N2) Resectable NSCLC, 30 (15/6/9)	1 cycle of atezolizumab (1200 mg) preoperatively	3 weeks after atezolizumab and within <15 days of that window	30 (100%)/30	24 days None delayed >15 days	NA/NA	trAEs Any Grade: 1/30 (3) trAEs ≥ G3: 0/30 (3)

trAEs = Treatment-Related; NR = Not Reached; NA = Not Available.

**Table 3 cancers-14-00741-t003:** Summary of combination neoadjuvant immunotherapy and immunochemotherapy trials with reported interim results.

Study	NCT Number	Population, *n* Enrolled	Study Arm	Control Group	Primary Endpoints	MPR and pCR Rates, *n/n* with Resection (%)	AEs or trAEs (%)
NEOSTAR (2021) [31] Random Phase II	03158129	Stage IA–IIIA Resectable NSCLC, 44	Two Parallel Arms: (1) 3 cycles of nivolumab preoperatively (2) 1 cycle of nivolumab + ipilimumab preoperatively	N/A	MPR	(1) MPR: 5/21 (24) Stage I/II/III: 2/2/1 pCR: 2/21 (10) (2) MPR: 8/16 (50) Stage I/II/III: 5/2/1 pCR: 6/16 (38)	(1) trAEs ≥ G3: 3/23 (13) (2) trAEs ≥ G3: 2/21 (10)
Reuss et al. (2020) [33] Phase Ib/II	02259621	Stage IB (≥4 cm)–IIIA Resectable NSCLC, 9	1 cycle of nivolumab + ipilimumab preoperatively with addition of 2 cycles of nivolaumb postoperatively	N/A	Safety and Feasibility	MPR: 0/6 (0) pCR: 2/6 (33) Stage I/II/III: 0/0/2	AEs Any Grade: 6/9 (67) AEs ≥ G3: 3/9 (33)
NADIM (2020) [34] Phase II	03081689	Stage IIIA Resectable NSCLC, 46	3 cycles of nivolumab, paclitaxel, and carboplatin preoperatively	N/A	PFS	MPR: 34/41 (83) Stage I/II/III: 0/1/33 pCR: 26/41 (63) Stage I/II/III: 0/1/25	AEs Any Grade: 43/46 (93) AEs ≥ G3: 14/46 (30)
Shu et al. (2020) [35] Phase II	02716038	Stage IB–IIIA Resectable NSCLC, 30	2 cycles of atezolizumab, nab-paclitaxel, and carboplatin preoperatively; if no progression seen, 2 more cycles given preoperatively	N/A	MPR	MPR: 17/26 (65) pCR: 10/26 (38)	trAEs Any Grade: 28/30 (93%) trAEs ≥ G3: 15/30 (50%)
Zinner et al. (2020) Phase II [36]	03366766	Stage IB (≥4 cm)–IIIA Resectable NSCLC, 13	3 cycles of nivolumab, cisplatin, and pemetrexed or gemcitabine preoperatively	N/A	MPR	MPR: 6/13 (85) pCR: 5/13 (39)	AEs ≥ G3: 2/13 (15)
Rothschild et al. (2021) [37] Phase II	02572843	Stage IIIA (N2) Resectable NSCLC, 68	3 cycles of cisplatin and docetaxel followed by 2 cycles of durvalumab preoperatively. Durvalumab continued for 1 year adjuvantly	N/A	1-Year EFS	MPR: 34/55 (62) pCR: 10/55 (18)	AEs Any Grade: 67/67 (100%) AEs ≥ G3: 59/67 (88)
CheckMate-816 (2021) [38,39] Random Phase III	02998528	Stage IB (≥4 cm)–IIIA Resectable NSCLC, 179 in each arm	3 cycles of nivolumab with platinum doublet preoperatively	3 cycles of platinum doublet preop-eratively	pCR and EFS	MPR Study: 66/179 (37) pCR Study Arm: 43/179 (24) MPR Control: 4/179 (2) pCR Control: 16/179 (9)	trAEs ≥ G3 Study: 60/179 (33.5) trAEs ≥ G3 Control: 66/179 (37)
Zhao et al. (2021) [40] Phase II	04304248	Stage IIIA-T3-4N2 IIIB Resectable NSCLC, 33	3 cycles of toripalimab, carboplatin, and pemetrexed or nab-paclitaxel preoperatively	N/A	MPR	MPR: 20/30 (66) pCR: 15/30 (50)	trAEs ≥ G3: 3/33 (9)
Shen et al. (2021) [41]	N/A	Stage IIB–IIIB Resectable NSCLC, 37	2 cycles of pembrolizumab, nab-paclitaxel, carboplatin pre-operatively	N/A	pCR	MPR: 24/37 (65) Stage I/II/III: 0/2/22 pCR: 17/37 (54%)	AEs ≥ G3: 5/37 (13.5)
Lei et al. (2020) [42] Random Phase III	04338620	Stage IIIA-IIIB(N2) Resectable NSCLC, 27 (14 in study arm) and 13 in control	3 cycles of camrelizumab, nab-paclitaxel, cisplatin preoperatively	3 cycles of nab-paclitaxel and cisiplatin preop-eratively	pCR	MPR: 6/7 (86) pCR: 4/7 (57)	Not reported in abstract
Tfayli et al. (2020) [43] Phase II	03480230	Stage IB (≥4 cm)–IIIA Resectable NSCLC, 15	4 cycles of avelumab with 3 cycles of cisplatin or carboplatin + gemcitabine or pemetrexed preoperatively	N/A	ORR	MPR: 3/11 (27) Stage I/II/III: 0/1/2 pCR: 1/11 (9) Stage I/II/III: 0/0/1	AEs ≥ G3: 4/15 (27)
Yang et al. (2018) [44] Phase II	01820754	Stage IB–IIIA Resectable NSCLC, 24	2 cycles of ipilimumab with 3 cycles of paclitaxel + cisplatin or carboplatin preoperatively	N/A	Surgical Outcomes. Safety	MPR: 2/13 (15) Stage I/II/III: 0/0/2 pCR: 2/13 (15) Stage I/II/III: 0/0/2	AEs Any Grade: 9/13 (69) AEs ≥ G3: 5/13 (38)

MPR = Major Pathologic Response; pCR = Pathologic Complete Response; AEs = Adverse Events; trAEs = Treatment-Related Adverse Events; EFS = Event Free Survival; ORR = Overall Response Rate; NR = Not Reached; N/A = Not Applicable; G3: grade 3. If there is no further breakdown into stages of NSCLC within the table, it means there is no corresponding reported data.

**Table 4 cancers-14-00741-t004:** Summary of Upcoming Neoadjuvant Radiotherapy with Neoadjuvant Immunotherapy or Chemotherapy Trials.

Study	NCT Number or EudraCT Number	Population, *n* Enrolled	Study Arm	Control Arm	Adjuvant Treatment	Primary Endpoints	Estimated Primary Completion Date
INCREASE Phase II	EudraCT number: 2019-003454-83	cT3-4, N0-1, M0 resectable or borderline resectable NSCLC, 29	Ipilimumab and nivolumab followed by nivolumab after 3 weeks plus platinum-doublet chemotherapy given concurrently with radiotherapy (50 Gy) over 5 weeks preoperatively	NA	NA	pCR and Safety	30 April 2024
N/A Phase II	03871153	Stage III (N2) resectable NSCLC, 25	Paclitaxel plus carboplatin plus durvalumab given concurrently with radiotherapy (45–61.2 Gy) over 5–6 weeks preoperatively	N/A	Durvalumab for 24 weeks	pCR	April 2022
N/A Phase II	03694236	Stage III resectable NSCLC, 39	Paclitaxel, carboplatin, and durvalumab given concurrently with radiotherapy (45 Gy) over 5 weeks preoperatively	N/A	N/A	pCR	May 2027
N/A Phase II	03237377	Stage III resectable NSCLC, 32	3 cycles of durvalumab with first cycle given concurrently with radiotherapy (45 Gy) over 5 weeks preoperatively vs. 3 cycles of durvalumab plus tremelimumab with first cycle given concurrently with radiotherapy (45 Gy) over 5 weeks preoperatively	N/A	N/A	Safety and Feas-ibility	September 2022
N/A, Phase I and Phase II	03446911	Stage I (T1cN0, T2aN0, T2bN0) peripherally located resectable NSCLC, 20	SABR preoperatively for phase 1; 2 cycles of pembrolizumab with first cycle given concurrently with SABR preoperatively for phase 2	N/A	N/A	Safety	May 2020

Trial information obtained from ClinicalTrials.gov. pCR = Pathologic Complete Response, N/A = Not Applicable.

**Table 5 cancers-14-00741-t005:** Summary of Upcoming Randomized Phase III Neoadjuvant Chemoimmunotherapy Trials.

Study	NCT Number	Population, *n* Enrolled	Study Arm	Control Arm	Adjuvant Treatment	Primary Endpoints	Estimated Primary Completion Date
AEGEAN	03800134	Stage IIA–IIIB Resectable NSCLC with PD-L1 TC ≥1%, 800/	4 cycles of durvalumab and platinum doublet pre-operatively	4 cycles of placebo and platinum doublet pre-operatively followed + placebo for 1 year post-operatively	Durvalumab for 48 weeks	MPR and EFS	30 April 2024
CheckMate- 77T	04025879	Stage IIA–IIIB (N2) Resectable NSCLC, 452	4 cycles of nivolumab and platinum doublet pre-operatively	4 cycles of placebo and platinum doublet pre-operatively + placebo for 1 year post-operatively	Nivolumab for 1 year	EFS	10 December 2023
CheckMate-816	02998528	Stage IB–IIIA Resectable NSCLC, 350	1 cycle of ipilimumab and 3 cycles of nivolumab pre-operatively vs. 3 cycles of nivolumab plus platinum doublet pre-operatively	3 cycles of platinum doublet pre-operatively	N/A	pCR and EFS	8 May 2023
Impower030	03456063	Stage II–IIIB (N2) Resectable NSCLC, 450	4 cycles of atezolizumab and platinum doublet pre-operatively	4 cycles of placebo and platinum doublet pre-operatively + surveillance post-operatively	Atezolizumab for 48 weeks	MPR and EFS	20 April 2025
KEYNOTE-671	03425643	Stage II–IIIB (N2) Resectable NSCLC, 786	4 cycles of pembro-lizumab and platinum doublet pre-operatively	4 cycles of placebo and platinum doublet pre-operatively + placebo for 39 weeks post-operatively	Pembrolizumab for 39 weeks	EFS and OS	20 January 2024

Trial information obtained from ClinicalTrials.gov. MPR = Major Pathologic Response; pCR = Pathologic Complete Response; EFS = Event Free Survival; OS = Overall Survival; N/A = Not Applicable.

## Data Availability

Not applicable.

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
