# Peer review of "Exploring the Evolving Scope of Neoadjuvant Immunotherapy in NSCLC"

_cancers, 2022, doi:10.3390/cancers14030741_

Round 1
Reviewer 1 Report
The reviewer thinks this is a well-written review article summarizing neoadjuvant immunotherapies in NSCLC. Several minor comments from the reviewer are listed below.
- Line 172, sintilimab would be a PD-1 inhibitor but not a PD-L1 inhibitor.
- Line 181, "sinitilimab" would be a typographical error. "i" is not necessary after "sin".
- In clinical trials listed in Tables 1 -5 (Tables 1 and 2 show the same trials), are there any inclusion criteria regarding the pre-treatment PD-L1 status? If some trials have such inclusion criteria, the reviewer suggests to add it in these Tables since this information is important when we discuss the efficacy data (especially for the MPR and pCR).
- The reviewer suggests to add a little bit more data about the MPR and pCR data of CM816 in the main text (Lines 313-318). ctDNA data is also reported in this trial (CM816), therefore, the reviewer thinks it is of worth adding some sentences in section 9 (Role of circulating tumor DNA and immune biomarkers in neoadjuvant immunotherapy assessments".
- In Table 5, there would be a small error in the "adjuvant treatment column" of AEGEAN trial.
Author Response
Dear Reviewer 1,
We first wanted to thank you for your time and effort in carefully reviewing our manuscript. Your comments and critiques are important aspects within the process of making the best manuscript we provide. We have scrupulously considered and addressed each of your critiques as below:
- We changed Line 172 to reflect that sintilimab is a PD-1 inhibitor and not a PD-L1 inhibitor
- We fixed the typographical error in line 181. Thank you for pointing this out.
- Upon review of the protocols of each trial analyzed in this review, only the AEGEAN trial had a pre-specified pre-treatment PD-L1 TC score ≥1% included in its inclusion criteria. Therefore, we placed this information in Table 5 within the column that described the population enrolled. All of the other trials did not specify that there had to be a specific pre-treatment PD-L1 status to be able to enroll.
- We added information about the MPR and pCR data of CM816 in lines 321-329 of revised manuscript. We additionally added information about ctDNA evaluation within this trial in lines 441-452 of the revised manuscript.
All of the above revisions were reviewed and approved by each author of the manuscript.
Thank you again for your consideration.
Sincerely,
Jun Zhang, MD, PhD
Associate Professor
Division of Medical Oncology, Department of Internal Medicine
University of Kansas Medical / Cancer Centers
Clinic: 2330 Shawnee Mission Pkwy #210, Westwood, KS 66205
Lab: 3005B Wahl Hall East, 3901 Rainbow Blvd, Kansas City, KS 66160 Telephone: (913) 588-6029
jzhang3@kumc.edu
Reviewer 2 Report
This is a well-written review of the current status of neoadjuvant immunotherapy trials including future directions in the field. After a concise, yet explicit introduction to neoadjuvant therapy and immunotherapy in oncology, the authors address individual trials, their findings and address future directions. They also discuss opportunities in the definition of trial endpoints and biomarker development.
The major concern is that the manuscript (MS) does not include the reference to the recent meta analysis by Cao, et. al. that summarizes neoadjuvant immunotherapy trials in NSCLC up to August 2021 (PMID: 34898553 Meta-Analysis of Neoadjuvant Immunotherapy for Patients with Resectable Non-Small Cell Lung Cancer. Cao C, Le A, Bott M, Yang CJ, Gossot D, Melfi F, Tian DH, Guo A. Curr Oncol. 2021 Nov 14;28(6):4686-4701. doi: 10.3390/curroncol28060395). This paper covers most of the trials included in the MS under review. Cao's paper should have been mentioned in the Discussion section and its conclusions should have been analyzed from the authors' perspective.
Also, Cao references some trials with a more up-to-date reference as the authors of the MS under review (e.g. reference 37. Rothschild, S et al, 2020 ASCO abstract in the MS vs. the same group's full paper in J. Clin. Oncol. 2021, 39, 2872–2880 in the Cao paper).
Author Response
Dear Reviewer 2,
We first wanted to thank you for your time and effort in carefully reviewing our manuscript. Your comments and critiques are important aspects within the process of making the best manuscript we provide. We have scrupulously considered and addressed each of your critiques as below:
- Thank you for pointing out our oversight in not including the Cao et al. meta-analysis. In lines 469-493 of our revised manuscript, we have included in the discussion section an overview of the meta-analysis as well as comments of what we thought were key points within this systematic review.
- We adjusted Reference 37 as suggested to reflect the more up-to-date reference as suggested. We also reviewed our other references to ensure they were up-to-date.
All of the above revisions were reviewed and approved by each author of the manuscript.
Thank you again for your consideration.
Sincerely,
Jun Zhang, MD, PhD
Associate Professor
Division of Medical Oncology, Department of Internal Medicine
University of Kansas Medical / Cancer Centers
Clinic: 2330 Shawnee Mission Pkwy #210, Westwood, KS 66205
Lab: 3005B Wahl Hall East, 3901 Rainbow Blvd, Kansas City, KS 66160 Telephone: (913) 588-6029
jzhang3@kumc.edu
Round 2
Reviewer 2 Report
I would like to thank the authors for responding to my comments adequately, I have no further suggestions.